# Hospital Malnutrition, a Call for Political Action: A Public Health and NutritionDay Perspective

**DOI:** 10.3390/jcm8122048

**Published:** 2019-11-22

**Authors:** Michael Hiesmayr, Silvia Tarantino, Sigrid Moick, Alessandro Laviano, Isabella Sulz, Mohamed Mouhieddine, Christian Schuh, Dorothee Volkert, Judit Simon, Karin Schindler

**Affiliations:** 1Division Cardio-thoracic and Vascular Anesthesia and Intensive Care, Medical University Vienna, 1090 Vienna, Austria; silvia.tarantino@meduniwien.ac.at (S.T.); sigrid.moick@gmail.com (S.M.); mohamed.mouhieddine@meduniwien.ac.at (M.M.); 2Center for Medical Statistics, Informatics and Intelligent Systems, Medical University Vienna, 1090 Vienna, Austria; isabella.sulz@meduniwien.ac.at (I.S.); christian.schuh@meduniwien.ac.at (C.S.); 3Department of Health Economics, Center for Public Health, Medical University Vienna, 1090 Vienna, Austria; judit.simon@meduniwien.ac.at; 4Department of Internal Medicine III, Medical University Vienna, 1090 Vienna, Austria; karin.schindler@meduniwien.ac.at; 5Department of Translational and Precision Medicine, Università degli Studi di Roma “La Sapienza”, 00185 Roma, Italy; alessandro.laviano@uniroma1.it; 6Institute for Biomedicine of Ageing, Friedrich-Alexander Universität Erlangen-Nürnberg, 90408 Nürnberg, Germany; dorothee.volkert@fau.de

**Keywords:** malnutrition, hospital, nutrition care, continuity of care, mortality, process indicators, benchmarking, disease related malnutrition.

## Abstract

Disease-related malnutrition (DRM) is prevalent in hospitals and is associated with increased care needs, prolonged hospital stay, delayed rehabilitation and death. Nutrition care process related activities such as screening, assessment and treatment has been advocated by scientific societies and patient organizations but implementation is variable. We analysed the cross-sectional nutritionDay database for prevalence of nutrition risk factors, care processes and outcome for medical, surgical, long-term care and other patients (*n =* 153,470). In 59,126 medical patients included between 2006 and 2015 the prevalence of recent weight loss (45%), history of decreased eating (48%) and low actual eating (53%) was more prevalent than low BMI (8%). Each of these risk factors was associated with a large increase in 30 days hospital mortality. A similar pattern is found in all four patient groups. Nutrition care processes increase slightly with the presence of risk factors but are never done in more than 50% of the patients. Only a third of patients not eating in hospital receive oral nutritional supplements or artificial nutrition. We suggest that political action should be taken to raise awareness and formal education on all aspects related to DRM for all stakeholders, to create and support responsibilities within hospitals, and to create adequate reimbursement schemes. Collection of routine and benchmarking data is crucial to tackle DRM.

## 1. Introduction

Disease-related malnutrition (DRM) is highly prevalent in hospitalized patients and associated with complications and poor outcome [1,2,3]. Malnutrition in hospitals originates from imbalances, either deficiencies or excesses, in nutrients intake compared with body needs. Nutritional status and needs may be modified acutely or chronically by the disease process itself. Further deterioration may occur due to hospitalization, thus making the population of hospitalized patients very different from the general population. Prolonged nutrients imbalance is associated with change in body mass index (BMI). Association between body mass index (BMI) and mortality is U-shaped in the general population and J-shaped in patients, especially with chronic diseases, meaning that mortality is higher if BMI is low and lower in patients with increased lean body mass and even obesity [4,5]. This observation called “obesity paradox” underscores the importance of a “good” nutrition status for patients with illnesses for short and long-term outcomes.

Malnutrition in hospitalized patients often addresses an evident poor nutritional status (low BMI and low muscle mass) whereas being at risk of malnutrition is derived from a set of risk factors typically associated with a loss in lean body mass persisting over a certain period of time [6]. Nutrition care in hospitals is a treatment for patients with malnutrition and a preventive intervention for patients at risk of malnutrition. Nevertheless, nutrition care is still an underrated field when compared to medical diagnostics procedures, or pharmacological and technological interventions in hospitals.

There are several reasons why nutrition care has received so little attention in acute care hospitals. There is a common knowledge deficit reflected in a lack of proper education in university curricula for healthcare professionals (doctors, nurses, care assistants) combined with patients and relatives giving low value to nutrition as part of a successful therapy of the primary disease. Since no immediate effects of nutrition can be expected during a short hospital length of stay, the attention for nutrition related issues is often low. Standard nutrition care processes such as screening, assessment, planning and monitoring together with documentation and continuity of care are not regular parts of care on all hospital wards [7]. Moreover, food provision and related tasks are not considered part of healthcare responsibilities. Food provision is not part of the medical budget but usually of the administrative budget of a hospital where cost reduction is not considered to influence directly patient care. Inadequate management of food provision might affect food quality, presentation [8] and composition and, subsequently, patient care. Food costs are to be added to the overall malnutrition related costs. Patients with malnutrition usually stay longer in hospitals, are more often re-hospitalized or transferred to long-term care [9,10,11,12]. Each of these, together with the reimbursement schemes, do create additional costs for the healthcare system. Insights on improved hospital nutrition care processes and reduced healthcare costs come from the Swiss EFFORT study.

In patients with risk of malnutrition, nutrition intake was improved with enriched meals or oral nutritional supplements, food intake was monitored and was associated with better outcomes and reduced healthcare costs [11].

These multiple barriers and the lack of proper attention to DRM and nutrition care in hospitals has been formally addressed at a political level by a resolution of the European Council in 2003 but was not followed by national regulatory actions [3].This lack of action led to several independent initiatives. The European Society for Clinical Nutrition and Metabolism (ESPEN) (www.espen.org) together with the Medical University of Vienna developed a tailored action to tackle malnutrition in hospitals and healthcare institutions [4]. The resulting project “nutritionDay” aimed to generate more awareness on DRM with a yearly one-day data collection on patient’s malnutrition risk factors, outcomes and quality indicators of nutrition care (www.nutritionday.org). In the nutritionDay analysis, several nutrition related risk factors, such as low BMI, recent weight loss and reduced food intake (in the week before nutritionDay or on nutritionDay itself), were found to be independently associated with death within 30 days in hospital [4]. In an another analysis that led to the development of the PANDORA score for prediction of death in hospital within 30 days after nutritionDay, decreased food intake was identified among the seven most important risk factors as it contributed 3–12 points out of a maximum of 75 points to the final score [5]. nutritionDay rapidly became a worldwide benchmarking tool for monitoring and improving nutrition care in hospital wards [13,14,15]. Patient related risk factors for decreased eating have a similar pattern in all world regions [16] and nutrition care processes implementation appears highly variable [7].

A multi-stakeholder initiative to promote screening for risk of disease-related malnutrition/undernutrition and implement nutritional care across Europe “Optimal Nutritional Care for All” has started an annual meeting with national nutrition societies and political representatives in 2008 [17]. The last meeting in Sintra, Portugal in 2018 involved representatives from 18 countries that had already joined the initiative and had the motto: “Optimal Nutrition Care across Europe: Fair Access and Shared Decision Making”. The decision was taken together with the European Patient Forum [18] to translate relevant guidelines into a lay version to increase the possibility for patients to take informed responsibility [19].

The human right for proper nutrition care was acknowledged in the Cartagena declaration [20] and signed on 3rd May 2019 by all presidents of the Latin American Federation of Nutritional Therapy, Clinical Nutrition (FELANPE). The declaration includes 13 principles, such as patient empowerment, dignity, ethical principles, justice and equity and urges the United Nations and the Human Rights Council to recognize the Right to Nutrition Care as a human right in line with the “Sustainable Development Goals” [21].

The worldwide Global Leadership Initiative on Malnutrition (GLIM) composed by the major nutrition societies aims at defining universally accepted criteria for DRM [22]. This process is still ongoing with planned steps of validation and regular updating. Current recommendations from scientific societies emphasize that it is mandatory to identify the malnourished as well as those at risk early during hospitalization to trigger proper treatment or a set of preventive measures [19]. The recommended three step process consists of screening, assessing and developing a nutrition care plan. A further important and less appreciated step is that nutrition care needs to be monitored and adapted to the patient’s changing condition. Finally, proper documentation and communication of a comprehensive care plan to the next sector, “extramural health care”, and the patients themselves are essential to ensure continuity of care. To be efficient, responsibilities need to be cleared delineated and all relevant stakeholder (Table 1) involved.

This study aims to determine in the medical patients of the nutritionDay database 2006–2018, first, the prevalence of simple nutrition related risk factors and their association with outcome and, second, to determine the routine use of recommended nutrition care procedures such as screening, nutrition intake monitoring and documentation in patients with and without risk factors.

Based on the findings we will propose several options for political action to tackle malnutrition in hospitals.

## 2. Experimental Section

The nutritionDay audit is a cross-sectional international data collection in hospitalized patients with 30 days in hospital outcome assessment. The nutritionDay project was approved by the Ethical Committee of the Medical University of Vienna (EK407/2005) and it has been amended annually. In accordance with national regulations, the project was also submitted to national or local ethical committees in each participating country. This trial was registered at clinicaltrials.gov as NCT02820246. We analysed the complete nutritionDay database 2006–2015 for the association between nutrition related risk factors and 30 days hospital mortality. We assessed the 2016–2018 nutritionDay database for nutrition care process indicators newly introduced in 2016.

We included all adult patients with the exception of women before or after giving birth. Patients were divided into four groups: group surgical (patients admitted to a surgical unit or waiting for surgery or after surgery on any medical/other ward), group medical (all patients admitted to general medical, cardio, gastro, hepatology, nephrology, infectiology or oncology), group long-term (geriatric or long-term care wards) and others (all other specialties such as ear-nose-throat, gynecology, obstetrics, trauma, orthopedics, etc.). Risk factors were age, gender, BMI, weight change in the last three months, decreased eating during the previous week and on nutritionDay, fluid status (as intravascular or tissue fluid overload or depletion as observed by a clinician), reduced mobility and whether patients have been admitted to an intensive care unit at any time before nutritionDay during this index admission. In addition we included in the multivariate analysis the diagnostic categories derived from the 17 ICD 10 top categories (brain and nerves, eye and ear, nose and throat, heart and circulation, lung, liver, gastrointestinal tract, kidney/urinary tract/female genital tract, endocrine system, skeleton/bone/muscle, blood/bone marrow, skin, ischemia, cancer, infection, pregnancy, others) and six comorbidities (diabetes, stroke, COPD, myocardial infarction, cardiac failure and others), each one used as a unique variable as condition being present versus not present. All factors were used as categorical variables and included a missing category according to the STROBE guidelines [23]. Reference categories were either the “normal” category or the group including the median or the largest subgroup as appropriate. Age was divided into eight groups spanning 10 years with 60–70 years as reference; BMI was divided into 6 WHO groups with normal BMI 18.5–25 used as a reference [24].

Sensitivity analysis included only patients from units fulfilling quality criteria, such as >60% recruitment of admitted patients and outcome reporting for >80% of included patients. In addition a second sensitivity analysis was done for the multivariate modelling of the association between risk indicators and outcome after exclusion of all cases with missing values. The sensitivity study population includes 82,993 patients compared with 153,470 in the full study population.

Descriptive statistics report frequencies and median with interquartile range. Comparison of proportions were done with the χ^2^ test and corrected for multiple testing. All associations between mortality and risk factors were done with logistic regression based on general linear models with units as clusters and weighting of patients to compensate for time-based bias from cross-sectional data acquisition [25]. In short, patients with a longer observed length of stay had less weight because they were more likely to be included in the study sample than short stay patients. We used univariate analysis for all risk factors and included all significant factors in the multivariate analysis. The effect of risk factors was also analysed within all four patient groups separately in the multivariate model (STATA 15.1, Statacorp, College Station, TX, USA). To show the extent of use of selected nutrition care processes, they are shown as percentages within different risk categories. Significant differences to each reference group were tested with a proportions test, comparisons were considered significant when *p* < 0.005 as we accounted for multiple tests for each reference category.

## 3. Results

Medical patients in the nutritionDay database (*n* = 59,126/153,470) (39%) represent the second largest group after surgical patients (*n* = 63,289/153,470) (41%). Patients were admitted in 61 countries and 19 countries recruited more than 1000 patients.

Medical patients were four years older than surgical patients (65.2 years SD 17.2 versus 61.2 years SD 18.0). The proportion of female was 48.3 in medical and 48.1 in surgical patients. Weight (71.3 kg SD 19.4 versus 71.9 kg SD 18.1) and height (166.1 cm SD 10.3 versus 166.5 cm SD 10.2) were similar in both groups. BMI was nearly identical (25.7 SD 6.3 versus 25.8 SD 5.8) (*p* = 0.02). Based on WHO categories (www.who.int) both groups have a similar proportion of obese with BMI > 30 (17.7% versus 17.4%) but the proportion with low BMI < 18.5 was significantly higher in medical than in surgical patients (7.5% versus 6%) (*p* < 0.0001).

### 3.1. Prevalence of Nutrition Risk Factors

Nutrition risk factors such as weight loss during the last three months (26,790, 45%), not eating normally in the previous week (28,950, 49%) and did not eat all food served on nutritionDay (30,965, 52%) were highly prevalent in the medical patients of the cohort 2006–2015 (Figure 1) and in the three other patient groups (Table 2).

There is some overlap between risk categories. Two third of patients (17,497/26,790) with weight loss reported not eating normal in the previous week and a similar proportion did not eat all served on nutritionDay (16,681/26,790).

Nearly 40% of patients (9551/24,679) that did eat normally in the previous week were eating less than all meal served on nutritionDay indicating a new nutrition risk associated with hospitalization.

In most patients (80%) all four nutrition risk factors, low BMI, weight loss, reduced eating during the previous week and on nutritionDay could be evaluated. 91% had no more than one risk factor missing. Only 32,216/140,418 (23%) of patients had no single nutrition related risk factor, 31% had one risk factor, 28% two risk factors and 16% had three risk factors.

### 3.2. Nutrition Care

#### 3.2.1. Food provision

Oral diet, either normal hospital food or special diet, was mostly used in medical patients (Table 3). Two third of oral diets were given as hospital food and one third as special diet. Oral nutritional supplements were given to 9.3% of patients whereas enteral nutrition was used in 7.1% and parenteral nutrition in 3.5% (Table 3). Surprisingly, the use of enteral or parenteral nutrition only increased by a factor of three between patients who reported having eaten their full meal and those who had eaten nothing. Two-thirds of patients reporting eating nothing were on oral diet. The use of enteral and parenteral nutrition was not differentiating much in patients who have eaten nothing regardless of being allowed to eat or not.

Percentages are indicated within each eating category. Multiple entries in the nutrition type for one patient are possible. Patients who ate nothing on nutritionDay were divided into those who did not eat even if they were allowed to (nothing_a) and those who did not eat because they were told not to eat by the doctor (nothing_na) as, for instance, before planned surgery or diagnostic tests. Combined enteral and parenteral nutrition was used in less than 2.5% of patients in any patient category and is not shown.

#### 3.2.2. Process indicators

A total of 1415 units reported in 2016–2018 about the screening tool utilised (Figure 2). The majority, used a formal tool such as NRS-2002 (Nutrition Risk Screening 2002), MUST (Malnutrition Universal Screening Tool), MST (Malnutrition Screening Tool), or a local tool. Only 10% of the units did not have a routine screening nor fixed screening criteria. Patients identified as malnourished were overall 3340/28,100 (11.9%), with the highest proportion identified with an informal tool 722/4793 (15.1%) or NRS-2002 1169/8453 (13.8%) followed by MUST 134/1210 (11.1%), MST 269/2269 (11.9%) and visual experience 197/1666 (11.8%) whereas the proportion of identified malnourished patients was lower with no implemented routine 204/2154 (9.5%). The proportion of patients identified at nutritional risk was overall 4944/28,100 (17.6%). Of those, the highest proportion, 22.4% was identified with an unspecified tool while only 14–16% of patients were identified at risk when no routine, only visual appearance or MST was used. Only 28.9% of malnourished patients had a BMI below 18.5, 49.8% a normal BMI, 21.4% were overweight or obese. Being identified as malnourished was associated with unintentional weight loss within the last three months in 85% of patients, with reduced nutrient intake before admission in 57% of patients and with not eating a full lunch in hospital in 65% of patients. Not being identified as malnourished or at risk of malnutrition was associated with unintentional weight loss within the last three months in 47% of patients, with reduced nutrient intake before admission in 28% of patients and with not eating a full lunch in hospital in 48% of patients. The proportion of patients with two or more nutrition risk factors was 50% in the 2016–2018 cohort.

Nutrition intake monitoring was more frequent in patients with unintentional weight loss than in patients with stable weight (52% versus 41%) (Figure 3). Still, nearly 50% of patients with a history of weight loss did not have their nutrition intake monitored while in hospital. Similarly, history of poor nutrition intake before admission triggered more frequent monitoring compared with history of normal eating but intake was never monitored in more than 50% of patients with a history of poor nutritional intake. Actual poor food intake did not have any effect on monitoring of food intake.

A nutrition expert was consulted more frequently when an unintentional weight loss was reported by patients or when food intake was reported as reduced before hospital admission (Figure 3). A nutrition expert was never consulted in more than 46% of patients even when risk factors were reported. Documentation of malnutrition in patient’s chart followed a similar pattern as consulting an expert and was always below 41%.

Low BMI was associated with a more frequent monitoring of intake (58% versus 47%) (*p* < 0.0001) and malnutrition reporting in patient record (52% versus 36%) (*p* < 0.0001) compared with normal BMI. In obese patients (BMI 30–35) we observed the lowest intake monitoring (40%) and malnutrition reporting (27%).

### 3.3. Outcome

Thirty days after nutritionDay we collected outcome data and found that 73% of the study patients had been discharged home, 11.8% had been transferred to another health care facility, hospital, long-term care or rehabilitation; 9.4% were still in the same hospital and 5.5% had died (Table 4). Similar results were observed in the sensitivity analysis. In medical patients, death was observed more than twice as frequently as in surgical patients (4.6% versus 1.7%, *p* < 0.0001) and was similar to the groups geriatrics and long-term care (4.6% versus 4.8%, *p* < 0.0001).

The group “other”, which includes patients from neurology, psychiatry and non-operative patients in gynecology and ENT, has a lower mortality (2.6% versus 4.6%) (*p* < 0.0001) a lower proportion discharged home *p* > 0.0001) at day 30 after nutritionDay than medical patients (Table 4).

Nutrition risk factors were associated with death in hospital within 30 days in a univariate analysis with odds ratios 2.6 CI 95 [2.2–3.0] for weight loss, 5.5 CI 95 [4.6–6.6] for eating less than a quarter of their meals in the previous week and 7.6 CI 95 [6.1–9.6] if eating nothing on nutritionDay despite being allowed to eat. The prevalence of individual risk categories for weight loss, decreased eating last week and on nutritionDay were associated with significantly worse outcomes (Figure 1), while increasing BMI was associated with decreasing odds ratios for death.

Other risk factors such as increasing age and male gender were also associated with poor outcome. The largest univariate association with hospital death was observed in patients with reduced mobility or those who were bedridden. Mortality within 30 days after nutritionDay increased with the number of nutrition risk factors present from 0.9% for no risk factor, 1.68% for one risk factor, 3.55% for two risk factors, 8.13% for three risk factors and 13% for four risk factors.

#### Multivariate Outcome Analysis

All risk factors that were significantly associated with 30 days hospital death in the univariate analysis were also significant in the multivariate analysis (Figure 4 and Figure 5) including all affected organs and comorbidities that could be estimated. Most estimates indicated a reduced strength of the association of the individual risk factors compared with univariate analysis, which indicates confounding. In the multivariate analysis, history of decreased eating before hospital admission was associated with higher odds ratio for death in surgical patients than in medical patients, weight loss was associated with similar odds ratio. History of a stay in intensive care during the hospitalization before nutritionDay was found to be significantly associated with death only in medical patients. “Eating less than normal in the previous week” is associated with death 30 days after nDay in medical, surgical and long-term care patients but not in the “other” group. Weight loss is only associated with poor outcome in medical and surgical patients. Higher BMI was associated with better outcome in medical, surgical and “other” patients, an observation called “reverse epidemiology” and low BMI with worse outcome but not in long-term care patients. Higher age is in all groups associated with worse outcome. Abnormal fluid status, identified as either overloaded or dehydrated, based on clinical judgment, was a robust risk indicator in all patient groups. Decreased food intake on nutritionDay was always associated with worse outcome. A clear difference between eating nothing while not being allowed (nil by mouth) or being allowed is only found in the surgical patient group whereas in all three other groups there was no clear difference. A very robust risk indicator is “reduced mobility” in all patient groups (Figure 4). Having had an ICU stay before nutritionDay was only associated with worse outcome in medical patients.

The association between affected organs as derived from ICD 10 categories is very different for medical and surgical patients (Figure 5). No comorbidities had a negative prognostic effect in the multivariate analysis.

## 4. Discussion

This analysis of medical/surgical patients from the nutritionDay cohort 2006–2015 showed that individual nutrition risk indicators such as low BMI, weight loss in last three months, poor eating before admission and low nutrient intake in hospital are highly prevalent in hospitalized patients and are associated with poor hospital outcome within 30 days after nutritionDay. Each additional risk indicator observed was associated with a nearly two-times higher mortality in the overall population.

All four risk indicators were independently associated with death in the hospital within 30 days after nutritionDay in the multivariate analysis (Figure 4) in the two largest groups medical and surgical patients. There is no clear indication that a single risk indicator can be ignored because the presence of each risk indicator is associated with a higher rate of dying by nearly a factor of two for these two groups. Surprisingly recent weight-loss and lower BMI was not associated with worse prognosis in long-term care whereas only weight-loss was not associated with worse outcome for the group of non-medical and non-surgical patients including neurological and psychiatric disorders.

Three risk indicators emerge similarly for all four patient groups, poor eating in hospital, reduced mobility and altered fluid status. All three had also being found as individual risk indicators in the PANDORA risk scoring system whereas recent weight loss and history of poor eating were not included [5]. These robust factors may also serve as indicators that may trigger more consistent observation and initiation of treatments that have proven to efficacious [26].

In recent years (2016–2018) healthcare teams identified only half of the patients as having nutrition related risk factors such as being malnourished or at risk of malnutrition in comparison to the nutritionDay indicators 8701/28,013 (31%) versus 21,070/28,100 (75%) such as unintentional weight loss, low BMI or decreased food intake. Thus, it appears that a large group of patients with nutrition related risk factors is not identified in standard care practice. This observed gap may originate from the different type of data collected from patients during routine clinical practice and the targeted data collection of the nutritionDay audit.

The efficiency of interventions may depend on the size of risk of the individual patient. It has recently been demonstrated that patients identified at high risk of malnutrition by the Nutritional Risk Screening (NRS-2002) risk system benefit from an integrated intervention [26]. It is unknown, however, whether the large group of patients with moderate risk, which account for almost half of the hospitalized patients on any given day, would also benefit from some targeted interventions. Interventions effective for the moderate risk group may be less complex and costly, and potentially integrated in to routine care processes, and could lead to better outcomes and reduced hospital length of stay and better patient outcomes.

### 4.1. Nutrition Care

Nutrition care processes such as screening, prescribing nutrition treatment, monitoring daily nutrition intake and recording of malnutrition status are applied in less than 50% of patients. These observations may again arise from identification issues, from lack of a systematic screening processes which then lead to a weak targeting even in the presence of nutrition risk factors. Educational gaps of healthcare professionals in the field of nutrition, as well as the importance given to nutrition care might also be considered crucial to fight DRM. Recent results from a large randomized trial [26] from a Swiss team could demonstrate that a systematic approach to patients with nutrition risk factors is associated with a decreased short and long-term mortality.

Our data suggest that a systematic care for patients with nutrition related risk factors in hospitals is currently missing. Reception and dissemination of current recommendations from major clinical nutrition societies such as ESPEN and ASPEN requires further efforts to reach all the involved stakeholders, not only those directly involved in nutrition care but also those in management and decision making.

### 4.2. Political Action Derived from Observations

A structured approach to tackle the challenges of disease related malnutrition needs to include all involved stakeholders to implement universal screening for risk factors, apply systematically supportive nutrition care, monitor the effect in individual patients and take measures to promote continuity of care after discharge from acute care hospitals.

### 4.3. Limitations of the Study

A limitation of such a cross-sectional international data collection may be that clinical observations and their evaluation are not matching the same clinical pattern. Participants are provided with a document explaining all collected variables in English and with questionnaires which were translated and checked by clinicians in order to achieve as much as possible consistency between answers. Great care was taken to utilize a simple and self-explanatory wording. This was also essential because the patient questionnaires are intended to be answered by patients independently of their education level. For such factors as fluid status, overloaded or dehydrated, we had to accept the data as clinically evaluated.

Such large databases with participating centres from more than 60 countries carry also the risk of missing data and non-homogeneous data reporting. Over the years we found stable proportions of missing data in the individual risk indicators. Surprisingly, this missingness does not appear to be at random as missing data were often associated with a poor outcome and not with an intermediate outcome from the various strata of a variable. We have addressed this issue by including missing categories in all variables as recommended by the STROBE statement [23] and have performed in addition a sensitivity analysis with complete cases without missing data that yielded quite similar results.

In addition, large databases may identify risk indicators with a minimal clinical relevance despite a large univariate effect. We have, thus, selected to present multivariate analysis where spurious associations are less likely.

A further limitation is that cross-sectional observational data from a convenience sample are not appropriate to determine a causal association between risk indicators and outcome. Nevertheless only observational studies can identify risk indicators because risk indicators cannot be randomised (Table 5).

## 5. Conclusions

In summary the analysis and dissemination of the nutritionDay data helps to increase awareness of nutrition related issues in hospitals and provides discussion for potential political actions. There are many options for political action. The key points appear to be the raised awareness about all the described aspects related to DRM to all stakeholders, as well as the proposal of easy and standardized nutrition care processes, defined responsibilities within hospitals, and the establishment of adequate reimbursement schemes. Collection of data is crucial to allow monitoring of DRM as well as food provision at all levels and to allow benchmarking and discussion within the teams. DRM is unlikely to disappear from hospitals because disease acts as a strong driver towards malnutrition. Appropriate nutrition care should be continued after discharge to prevent further deterioration in patient’s autonomy, quality of life and poor outcome. A systematic approach to DRM in hospitals and an adequate continuity of care may lead to better outcomes [11,12,26]. nutritionDay in this regard has served and it will keep serving as a tool to monitor changes in clinical practice and associated outcome.

## Figures and Tables

**Figure 1 jcm-08-02048-f001:**
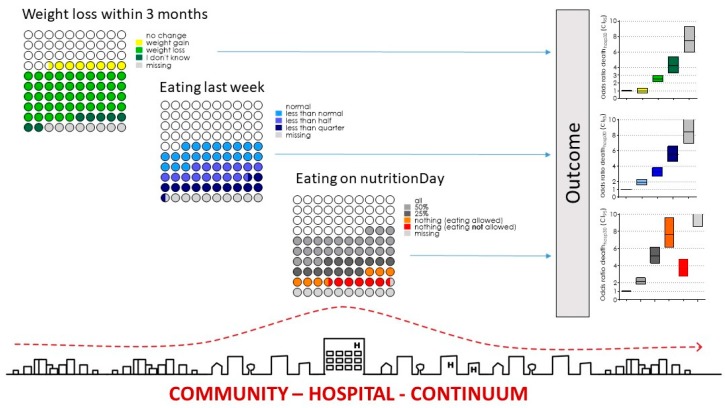
Prevalence of risk factors and association with odds ratio for death in hospital within 30 days after nutritionDay in medical patients. Prevalence is indicated by dots. Each dot represents 1% of the total population. All risk indicators are collected on one single day, the nutritionDay 2006–2015. Odds ratio are indicated with 95% confidence intervals and colours according to risk indicator categories Graph of Community–Hospital–Continuum from Magdalena Maierhofer’s architectural diploma thesis: A Hospital is not a Tree (2016).

**Figure 2 jcm-08-02048-f002:**
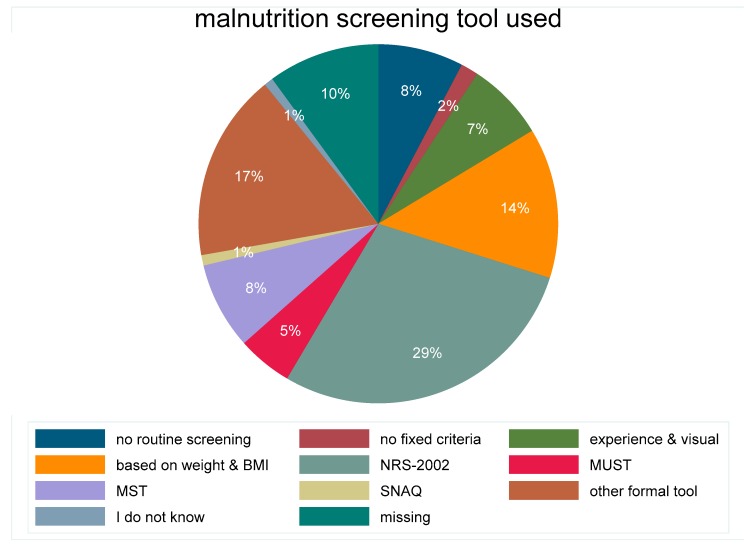
Proportion of different methods/approaches used for malnutrition screening in 1415 units from 46 countries in the nutritionDay cohort 2016–2018. NRS-2002 (nutrition risk screening 2002); MUST (Malnutrition Universal Screening Tool); MST (Malnutrition Screening Tool); SNAQ (Short Nutritional Assessment Questionnaire).

**Figure 3 jcm-08-02048-f003:**
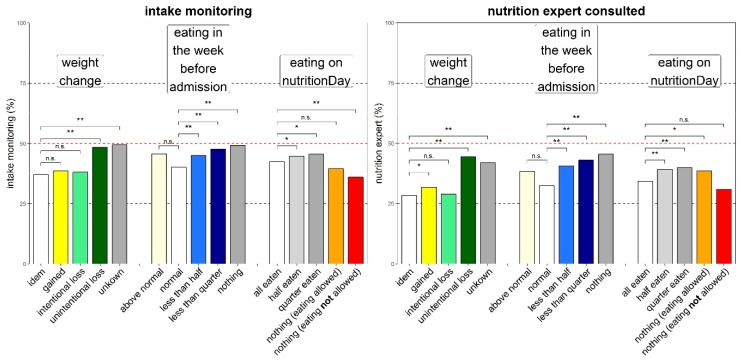
Nutrition care process indicators versus three nutrition associated risk factors. Daily nutrition intake monitoring (**left**) and nutrition expert consulted (**right**). Bars indicate percentage answering “yes”, significant differences to each reference group are shown with * *p* < 0.005 and ** *p* < 0.00001, ‘n.s.’ indicates no significant difference. Missing values were <7.5% in all subcategories. Colour coding similar to Figure 1 (the empty bar is the reference).

**Figure 4 jcm-08-02048-f004:**
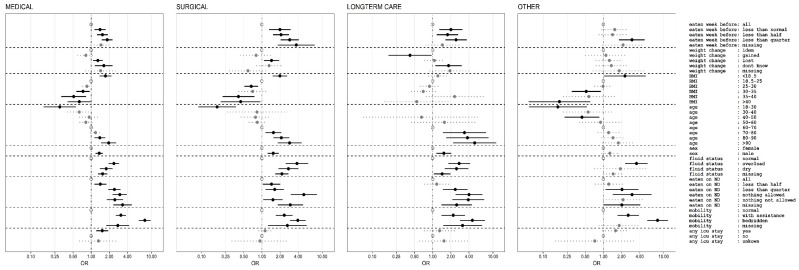
Multivariate analysis of association between demographic and nutrition related risk factors and death in hospital within 30 days after nutritionDay for medical, surgical, long-term care and other patient groups with the general linear model for logistic regression with wards as clusters and weighting of individual patients for sampling probability [25] and including all diagnostic categories and comorbidities (see Figure 5). Odds ratios (OR) with 95% confidence intervals indicated by horizontal line. Reference categories are indicated by an open symbol. Missing values are included in the model as individual categories.

**Figure 5 jcm-08-02048-f005:**
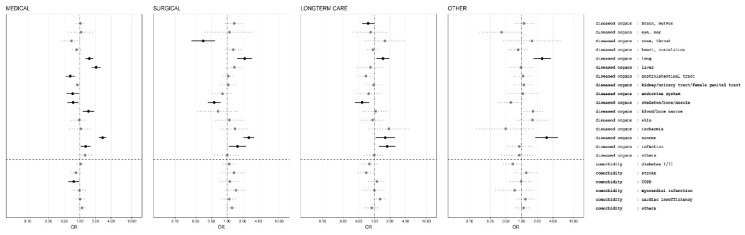
Multivariate analysis of association between organ related disease categories from ICD 10 as well as comorbidities and death in hospital within 30 days after nutritionDay for medical, surgical, long-term care and other patient groups with the general linear model for logistic regression with wards as clusters and weighting of individual patients for sampling probability [25] and including demographic and nutrition related risk factors (see Figure 4). Odds ratios (OR) with 95% confidence intervals indicated by horizontal line. Multiple entries are possible. Missing values are included in the model as individual categories.

**Table 1 jcm-08-02048-t001:** List and role of important stakeholders.

1. Within the hospital		
	a. Patients and their relatives	
	b. Care persons	
	i. Nurses	Screening, diet ordering, documentation
	ii. Physicians	Assessment, ordering, documentation, information
	iii. Dieticians	Assessment, documentation
	iv. Physiotherapists	Effect monitoring
	v. Speech Therapists	Swallowing disorders
	vi. Pharmacists	Clinical nutrition supply and counselling
	c. Kitchen/Catering services	
	i. Administrators	Budget
	ii. Chefs	Standards, variety, quality control
	iii. Kitchen aids	Presentation
	iv. Delivering staff	Monitoring
	d. Hospital administration	Budget, planning, controlling
2. Outside the hospital		
	a. Patients and relatives	
	b. Extramural medical services/family medicine/primary health care centres	
	c. Extramural care services/mobile nursing	
	d. Services for disabled and dependent persons	
	e. Local food producers	
	f. Medical food producing industries	
3. Scientific societies and stakeholder associations		
	a. Medical	Guidelines, standards
	b. Nursing	Guidelines, standards
	c. Dietician	Guidelines, standards
	d. Nutrition science	Research, standards
	e. Patient organizations	Guidelines
4. Policy maker		
	a. Health care system	Reimbursement
	b. Social affairs	Equity
	c. Agriculture	Local production integration
	d. Environmental affairs	Sustainable planning, waste prevention
5. Payers		
	a. Reimbursement of the nutrition care process in the whole health care system	
	b. Public procurement of food supply and services	

**Table 2 jcm-08-02048-t002:** Demographic characteristics and prevalence of nutrition risk factors of patients in the nutritionDay cohort 2006–2015, in the four studied groups.

	Medical		Surgical		Long-Term Care		Others	
Characteristics	*n* (%)	Mean ± SD	*n* (%)	Mean ± SD	*n* (%)	Mean ± SD	*n* (%)	Mean ± SD
Age (year)	59,046	65.1 ± 17.2		61.2 ± 18.0		80.7 ± 12.4		61.2 ± 18.8
Gender (female)	28,535 (49%)		30,445 (49%)		6937 (62%)		9992 (51%)	
Weight (kg)	52,735 (89%)	71.3 ± 19.4		71.9 ± 18.1		66.4 ± 16.9		71.8 ± 18.3
Height (cm)	52,735	166.1 ± 10.3		166.6 ± 10.2		163.0 ± 9.5		166.7 ± 10.4
BMI * (kg·cm^−2^)	52,735	25.7 ± 6.3		25.8 ± 5.8		24.9 ± 5.8		25.7 ± 5.9
**Weight change within three months**								
Unchanged *	19,139 (32%)		25,164 (40%)		3153 (28%)		7338 (37%)	
Increase	4335 (7.3%)		4997 (8%)		671 (6%)		1990 (10%)	
Loss	26,790 (45%)		24,928 (39%)		4989 (44%)		7356 (37%)	
Do not know	4020 (6.8%)		3679 (6%)		1339 (12%)		1205 (6%)	
Missing	4842 (8.2%)		4521 (7%)		1127 (10%)		1887 (10%)	
**Eating last week**								
Normal *	24,679 (42%)		29,898 (47%)		4731 (42%)		9973 (50%)	
Less than normal	12,613 (21%)		12,618 (20%)		2526 (22%)		4047 (20%)	
Less than a half	8979 (15%)		7894 (12%)		1628 (14%)		2262 (11%)	
Less than a quarter	7358 (12%)		7691 (12%)		1076 (9%)		1433 (7%)	
Missing	5497 (9.3%)		5188 (8%)		1318 (12%)		2061 (10%)	
**Eating on nutritionDay**								
All *	22,046 (37%)		22,232 (35%)		4131 (37%)		8496 (43%)	
Half	15,327 (26%)		15,141 (24%)		3363 (30%)		5054 (26%)	
Quarter	8256 (14%)		7262 (11%)		1592 (14%)		2206 (11%)	
Nothing (eating allowed)	3696 (6.3%)		3666 (6%)		698 (6%)		927 (5%)	
Nothing (eating not allowed)	3686 (6.2%)		8717 (14%)		372 (3%)		875 (4%)	
Missing	6115 (10%)		6271 (10%)		1123 (10%)		2218 (11%)	
**Mobility on nutritionDay**								
Normal	35,846 (61%)		37,439 (59%)		3731 (33%)		12,227 (62%)	
With help	12,299 (21%)		14,110 (22%)		4557 (40%)		3838 (19%)	
Bedridden	5587 (9.4%)		6832 (11%)		1841 (16%)		1732 (9%)	
Missing	5394 (9.1%)		4908 (8%)		1150 (10%)		1979 (10%)	
**Fluid status on nutritionDay**								
Normal *	28,499 (48%)		3619 (6%)		1147 (10%)		1284 (6%)	
Overload	6214 (11%)		33,636 (53%)		5836 (51%)		1142 (58%)	
Deficit	3267 (6%)		2723 (4%)		985 (9%)		886 (4%)	
Missing	21,146 (36%)		23,311 (37%)		3311 (29%)		6178 (31%)	
**Any Intensive Care Stay before nutritionDay**	4143 (7.0%)		10,465 (17%)		564 (5%)		1464 (8%)	
**Medical specialty**								
General internal medicine	29,173 (49%)		3958 (6%)					
Oncology	11,412 (19%)		1953 (3%)					
Gastroenterology/Hepatology	9744 (16%)		1350 (2%)					
Cardiology	5401 (9.1%)		1405 (2%)					
Nephrology	1785 (3.0%)		370 (1%)					
Infectiology	1611 (2.7%)		149 (0%)					
Neurology			592 (1%)				4442 (22%)	
Psychiatry			17 (0%)				1409 (7%)	
ENT			2195 (3%)				1272 (6%)	
General surgery			28,310 (45%)					
Cardiothorcic surgery			2013 (3%)					
Orthopaedic surgery			7803 (12%)					
Trauma			2160 (3%)					
Neurosurgery			1717 (3%)					
Gynecology			1198 (2%)				1151(6%)	
Long-term care			526 (1%)		9885 (88%)			
Other			5955 (9%)				11,401(58%)	
Pediatrics			46 (0%)				101 (1%)	
Geriatrics			1572 (2%)		1785 (12%)			

* indicates the reference categories.

**Table 3 jcm-08-02048-t003:** Nutrition care versus amount eaten on nutritionDay according to the patients in the four studied groups.

		Oral	ONS	EN	PN	Othercomb
Medical	all	19,484 (88.4%)	1651 (7.5%)	1154 (5.2%)	289 (1.3%)	794 (3.6%)
*n =* 59,126	half	13,657 (89.1%)	1560 (10.2%)	770 (5%)	311 (2%)	544 (3.5%)
	quarter	7176 (87%)	1132 (13.7%)	426 (5.2%)	354 (4.3%)	332 (4%)
	nothing_a	2760 (74.7%)	448 (12.1%)	428 (11.6%)	306 (8.3%)	278 (7.5%)
	nothing_na	2391 (65%)	226 (6.1%)	329 (8.9%)	417 (11.3%)	521 (14.1%)
	missing	3377 (55.2%)	462 (7.6%)	1083 (17.7%)	381 (6.2%)	323 (5.3%)
	Total	48,845 (82.6%)	5479 (9.3%)	4190 (7.1%)	2058 (3.5%)	2792 (4.7%)
Surgical	all	19,368 (87.1%)	1468 (6.6%)	1286 (5.8%)	429 (1.9%)	1184 (5.3%)
*n =* 63,289	half	13,106 (86.6%)	1294 (8.5%)	913 (6%)	506 (3.3%)	906 (6%)
	quarter	6066 (83.5%)	779 (10.7%)	472 (6.5%)	381 (5.2%)	536 (7.4%)
	nothing_a	2384 (65%)	324 (8.8%)	448 (12.2%)	415 (11.3%)	573 (15.6%)
	nothing_na	4611 (53%)	301 (3.5%)	810 (9.3%)	1535 (17.6%)	1906 (21.9%)
	missing	3596 (57.3%)	392 (6.3%)	819 (13.1%)	605 (9.6%)	689 (11%)
	Total	49,131 (77.6%)	4558 (7.2%)	4748 (7.5%)	3871 (6.1%)	5794 (9.2%)
Longterm	all	3480 (84.3%)	824 (19.9%)	370 (9%)	15 (0.4%)	135 (3.3%)
*n =* 11,279	half	2858 (85%)	790 (23.5%)	268 (8%)	29 (0.9%)	113 (3.4%)
	quarter	1312 (82.4%)	482 (30.3%)	130 (8.2%)	28 (1.8%)	70 (4.4%)
	nothing_a	473 (67.8%)	225 (32.2%)	111 (15.9%)	41 (5.9%)	53 (7.6%)
	nothing_na	200 (53.8%)	75 (20.2%)	85 (22.8%)	46 (12.4%)	43 (11.6%)
	missing	672 (59.8%)	177 (15.8%)	247 (22%)	36 (3.2%)	45 (4%)
	Total	8995 (79.7%)	2573 (22.8%)	1211 (10.7%)	195 (1.7%)	459 (4.1%)
Others	all	7435 (87.5%)	509 (6%)	672 (7.9%)	70 (0.8%)	392 (4.6%)
*n =* 19,776	half	4398 (87%)	380 (7.5%)	403 (8%)	84 (1.7%)	258 (5.1%)
	quarter	1885 (85.4%)	267 (12.1%)	147 (6.7%)	78 (3.5%)	118 (5.3%)
	nothing_a	628 (67.7%)	84 (9.1%)	160 (17.3%)	68 (7.3%)	90 (9.7%)
	nothing_na	544 (62.2%)	49 (5.6%)	100 (11.4%)	101 (11.5%)	131 (15%)
	missing	1214 (54.7%)	124 (5.6%)	355 (16%)	91 (4.1%)	128 (5.8%)
	Total	16,104 (81.4%)	1413 (7.1%)	1837 (9.3%)	492 (2.5%)	1117 (5.6%)

**Table 4 jcm-08-02048-t004:** Outcomes in hospital within 30 days after nutritionDay in the four patient groups.

Outcome	Surgery	Medical	Longterm	Other
in hospital	5740 (9.1%)	4639 (7.8%)	1560 (13.8%)	2303 (11.6%)
transfer other hospital	1500 (2.4%)	1343 (2.3%)	290 (2.6%)	390 (2%)
transfer longterm care	1424 (2.2%)	2109 (3.6%)	1471 (13%)	574 (2.9%)
transfer rehabilitation	1967 (3.1%)	1341 (2.3%)	441 (3.9%)	535 (2.7%)
discharge home	39,705 (62.7%)	36,439 (61.6%)	4710 (41.8%)	11,049 (55.9%)
death within 30 days	1053 (1.7%)	2721 (4.6%)	541 (4.8%)	512 (2.6%)
other destination	948 (1.5%)	1033 (1.7%)	322 (2.9%)	376 (1.9%)
missing	10,952 (17.3%)	9501 (16.1%)	1944 (17.2%)	4037 (20.4%)

**Table 5 jcm-08-02048-t005:** Problem areas and suggested options for political action.

Problem Area	Political Action
Education of all healthcare professionals directly involved in patient care in disease related malnutrition and nutrition care insufficient.	Mandatory inclusion of disease related malnutrition and nutrition care processes in curriculum for nurses, doctors, dieticians, etc.
Limited awareness of the importance of nutrition in disease states in the public especially the population at risk.	National nutrition care campaigns targeting the general population, residents of nursing homes and also targeted nutrition campaigns run through general practitioners. Availability of an education platform for patients and families.
Nomination of responsible person or team for patient nutrition care missing. No monitoring of nutrition care processes part of hospital quality control.	Mandatory designation of a nutrition team/responsible person in each hospital with a threefold responsibility: coordination of expertise, definition of processes and regular benchmarking of applications of processes through initiatives like nutritionDay, the Dutch nutrition benchmarking program, the British malnutrition awareness week and the analysis of electronic patients records.
Inconsistent screening and collection of data. Missing documentation of nutrition risk factors and communication of nutrition status and care at discharge to the next sector.	Mandatory inclusion of data in a nutrition care benchmarking program. Definition and inclusion of mandatory harmonized fields for a systematic collection and documentation of nutrition risks factors and nutrition care processes in the electronic patient record. Inclusion of planned nutrition treatment recorded in patient’s discharge letter/information to patients and relatives.
Missing patients and families empowerment due to insufficient communication of nutrition status and care to the patients and their families.	Mandatory monitoring of communication processes in quality assurance programs.
Lack of a harmonized reimbursement schemes for nutrition related processes such as screening, assessment and treatment such as oral nutritional supplements, enteral or parenteral nutrition.	Clear reimbursement schemes.
Missing a partnership for hospital food provision and of a positive image for hospital food.	Creation of a public best practice platform for food provision in hospitals. Supported use of local food in hospital kitchen for the creation of wealth not only for the community using the hospital but also for the local community.

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
