# Peer review of "Hospital Malnutrition, a Call for Political Action: A Public Health and NutritionDay Perspective"

_jcm, 2019, doi:10.3390/jcm8122048_

Round 1

Reviewer 1 Report

This paper reports on disease-related malnutrition using the Nutrition Day day database. It is timely that this topic is receiving attention in the literature.

This paper requires further editing:

Page 2, line 71:It is unclear how this sentence links to the previous paragraph. Consider rewording for clarification.

Line 76-87:One sentence is not a paragraph. Link line 79-87 into this paragraph.

Line 92:....that had already joined ...

Line 94: ....into a lay version

Line 95: ..for patients to take informed responsibility.

Line 96: The human right..

Page 3 line 105-112: Include this in the previous paragraph

Line 106: Omit "in order" to trigger- in order is superfluous

Line 111: Ditto

Page 4, line 152: Clumsy wording in opening sentence- remove for 

Line 160: should it read: with a 30 day?

Line 177: suggest: with 60-70 years as reference

Line 179: ....quality criteria, such as....

Line 182: statistics report frequencies

Page 6, line 209-210: found in the 2016-2015 cohort ... and 2016 ad 2010 cohort

Line 211-213: expression- should it be?: not eating normally and did not eat all food served on nutritionDay

Line 221: Is, "On the other side"... Is this an appropriate expression here?

Line 222: meals

Line 225: Avoid starting a sentence with a percentage: suggest: Most patients (91%)...

Page 7, Line 228-9: This sentence belongs in Methods

Line 230: risk of becoming....

Line 232: BMI, 47% of patients had a normal BMI and 20% of patients were overweight or obese, while 6% were missing their BMI classification. 

Line 236: Two thirds of oral diets were given as hospital food or ...

Line 240: ...who had eaten nothing. Final sentence in paragraph is clumsy- reword.

Table 2: category headings, please clarify "Nothing eating allowed" and  "Nothing eating not allowed". These categories do ot make sense. Add data to final category ie missing data.

Line 248: :  nutrition treatment was provided

Line 249: Screening for ??? was done in....

Page 8, Line 251: ...69% of malnourished patients, 59% of at risk patients and 39% of well-nourished patients.

Line 252: Ditto

Line 269: Poor intake of ??? did not... of monitoring ??? intake

Page 9, line: 290 less than a quarter (of what?) .. in the ....

Line 291 The prevalence... worse outcomes, while increasing BMI was ...

Line 295: Other risk factors, such as ...

Line 297: ..or those who were bed ridden.

Page 10, Line 298 Include in previous paragraph

Line 303: you have used 30 days hospital death- requires further explanation ie do you mean death in hospital within 30 days as indicated elsewhere?

Line 306: analysis, which indicates confounding

Line 311: ...with the general linear...

Line 314: ,,,without a horizontal bar.

Line 321: You could use post nutritonDay throughout instead of after

Page 11, Line 326: Clumsy expression, flip the sentence, ie In recent years, health care teams identified ...

Line 334: Have you written out this abbreviation in full previously?

Line 338-9: ... and potentially integrated into routine care practices and lead to reduced hospital length of stay, and better patient outcomes.

Line 343 omit the ie from lack of a ....

Lines 345-6 Also used twice in 1 sentence

Line 351: ASPERN requires further efforts

Line 358: what does proper mean here? Suggest replacing with a more appropriate word or expand to clarify.

Line 361: Do you mean nutrients? Perhaps foods?

Page 12, final box in LHC Missing a partnership for ... 

RHC, box 2: ....homes and also target... Final sentence: ....education platform ... 

RHC, final box: Support for use of? 

Page 13, line 376: In summary the analysis... and provides discussion for potential

Line 379: ...DRM to all...

Line 383: DRM is unlikely to disappear from hospital related disease acting as a strong driver of malnutrition.

Line 384: Appropriate nutrition care should be ...

Author Response

We thank Reviewer 1 very much for helping us improving the clarity of the text, and the use of the English language. Almost all the revisions suggested where corrected accordingly.

Page 2, line 71: It is unclear how this sentence links to the previous paragraph. Consider rewording for clarification.

We merged a revised version of this sentence into the previous paragraph (line 67) where we describe the food provision task in hospital not only in terms of costs but also of quality, presentation and consequently patient care. Suggestions from reviewer were taken. Sentence was rephrased with:

Inadequate management of food provision might effect food quality, its presentation [8] and composition and subsequently patient care. Food costs are to be added to the…

Line 76-87:One sentence is not a paragraph. Link line 79-87 into this paragraph.

Suggestions from reviewer were taken. The sentence was linked to previous paragraph.

Line 92:....that had already joined ...

Suggestions from reviewer were taken. The grammatical mistake was corrected.

Line 94: ....into a lay version

Suggestions from reviewer were taken. The grammatical mistake was corrected.

Line 95: ..for patients to take informed responsibility.

Suggestions from reviewer were taken. The grammatical mistake was corrected.

Line 96: The human right..

Suggestions from reviewer were taken. The grammatical mistake was corrected.

Page 3 line 105-112: Include this in the previous paragraph

Suggestions from reviewer were taken. The sentences were included to previous paragraph.

Line 106: Omit "in order" to trigger- in order is superfluous

Suggestions from reviewer were taken. The sentence was corrected accordantly.

Line 111: Ditto

Suggestions from reviewer were taken. The sentence was corrected accordantly. “In order” was also here superfluous.

Page 4, line 152: Clumsy wording in opening sentence- remove for 

Suggestions from reviewer were taken. The sentence was corrected with:

This study aims to determine in the medical patients of the nutritionDay database 2006-2018 first the prevalence of simple nutrition related risk factors and its association with outcome and second to determine the routine use of recommended nutrition care procedures such as screening, nutrition intake monitoring and documentation in patients with and without risk factors.

Line 160: should it read: with a 30 day?

Suggestions from reviewer were taken. The grammatical mistake was corrected to 30 days

Line 177: suggest: with 60-70 years as reference

Suggestions from reviewer were taken. The sentence was corrected accordantly.

Line 179: ....quality criteria, such as....

The formatting suggestion was taken.

Line 182: statistics report frequencies

Suggestions from reviewer were taken. The sentence was corrected accordantly.

Page 6, line 209-210: found in the 2016-2015 cohort ... and 2016 ad 2010 cohort

Suggestions from reviewer were taken. The sentence was corrected accordantly.

Line 211-213: expression- should it be?: not eating normally and did not eat all food served on nutritionDay

Suggestions from reviewer were taken. The sentence was corrected accordantly.

Nutrition risk factors such as weight loss during the last 3 months (26790, 45%), not eating normally in the previous week (28950, 49%) and did not eat all food served on nutritionDay (30965, 52%) were highly prevalent in the medical patients of the cohort 2006-2015 and 2016-2018

Line 221: Is, "On the other side"... Is this an appropriate expression here?

Suggestions from reviewer were taken. The sentence was corrected with:

Nearly 40% of patients (9551/24679) eating normally in the previous week that were eating less than all meal served on nutritionDay indicating a new nutrition risk associated with hospitalization.

Line 222: meals

Suggestion from reviewer was not taken. nutritionDay is usually run after lunch, data on the food intake are collected just after that meal.

Line 225: Avoid starting a sentence with a percentage: suggest: Most patients (91%)...

Suggestions from reviewer were taken. The sentence was corrected accordantly.

Page 7, Line 228-9: This sentence belongs in Methods

Suggestions from reviewer were taken. The sentence was deleted from results section.

Line 230: risk of becoming....

Suggestions from reviewer were taken. The sentence was corrected accordantly.

Line 232: BMI, 47% of patients had a normal BMI and 20% of patients were overweight or obese, while 6% were missing their BMI classification. 

Suggestions from reviewer were taken. The sentence was corrected accordantly.

Line 236: Two thirds of oral diets were given as hospital food or ...

Suggestions from reviewer were taken. The sentence was corrected accordantly.

Line 240: ...who had eaten nothing. Final sentence in paragraph is clumsy- reword.

Suggestions from reviewer were taken. Last sentence of paragraph was rephrased with:

The use of enteral and parenteral nutrition was not differentiating much in patients who have eaten nothing regardless of being allowed or not to eat.

Table 2: category headings, please clarify "Nothing eating allowed" and  "Nothing eating not allowed". These categories do ot make sense. Add data to final category ie missing data.

Suggestions from reviewer were taken. We added a clarification as table footer:

 Patients who ate nothing on nutritionDay were distinguished in the ones who did not eat even if they were allowed to (nothing-eating allowed) and in the ones who did not eat because they were told not to by the doctor (nothing-eating not allowed) as it occurs for instance before a planned surgery.

 The suggestion Missing data was taken.

Line 248: :  nutrition treatment was provided

Suggestions from reviewer were taken. The sentence was corrected accordantly.

Line 249: Screening for ??? was done in....

Suggestions from reviewer were taken. Sentence was rephrased with:

Screening for malnutrition was done in 66% of malnourished patients, in 55% of at risk patients and 26% of the well-nourished patients.

Page 8, Line 251: ...69% of malnourished patients, 59% of at risk patients and 39% of well-nourished patients.

Suggestions from reviewer were taken. Sentence was rephrased with:

Nutrition intake was monitored in 69% of malnourished patients, 59% of at risk patients and 39% of well-nourished patients. Malnutrition status was recorded in the patient chart of 77% of malnourished patients, 54% of at risk patients and 21% of well- nourished patients.

Line 252: Ditto

Suggestions from reviewer were taken. Sentence was rephrased with:

Malnutrition status was recorded in the patient chart of 77% of malnourished patients, 54% of at risk patients and 21% of well- nourished patients.

Line 269: Poor intake of ??? did not... of monitoring ??? intake

Suggestions from reviewer were taken. Sentence was rephrased with:

Actual poor food intake did not have any effect on monitoring of food intake

Page 9, line: 290 less than a quarter (of what?) .. in the ....

 Suggestions from reviewer were taken. The sentence was corrected accordantly.

Nutrition risk factors were associated with death in hospital within 30 days in a univariate analysis with odds ratios 2.6 CI 95 [2.2-3.0] for weight loss, 5.5 CI 95 [4.6-6.6] for eating less than a quarter of their meals in the previous week and 7.6 CI 95 [6.1-9.6] if eating nothing on nutritionDay despite being allowed to eat.

Line 291 The prevalence... worse outcomes, while increasing BMI was ...

Suggestions from reviewer were taken. Sentence was rephrased with:

The prevalence of individual risk categories for weight loss, decreased eating last week and on nutritionDay were associated with significantly worse outcomes (Figure 1),while increasing BMI was associated with decreasing odds ratios for death.

Line 295: Other risk factors, such as ...

Suggestions from reviewer were taken. Sentence was rephrased with:

Line 297: ..or those who were bed ridden.

Suggestions from reviewer were taken. Sentence was rephrased accordantly.

Page 10, Line 298 Include in previous paragraph

Suggestions from reviewer were taken. Sentence was included in previous paragraph.

Line 303: you have used 30 days hospital death- requires further explanation ie do you mean death in hospital within 30 days as indicated elsewhere?

Suggestions from reviewer were taken. Sentence was rephrased with:

Line 306: analysis, which indicates confounding

Suggestions from reviewer were taken. Sentence was rephrased accordantly.

Line 311: ...with the general linear...

Suggestions from reviewer were taken. Sentence was rephrased accordantly.

Line 314: ,,,without a horizontal bar.

Suggestions from reviewer were taken. Sentence was rephrased accordantly.

Line 321: You could use post nutritonDay throughout instead of after.

Thank you for the suggestion, we still used the expression “after nutrionDay”.

Page 11, Line 326: Clumsy expression, flip the sentence, ie In recent years, health care teams identified ...

 Suggestions from reviewer were taken. Sentence was rephrased accordantly.

Line 334: Have you written out this abbreviation in full previously?

Suggestions from reviewer were taken. Abbreviation was explained:

The efficiency of interventions may depend on the size of risk of the individual patient. It has recently been demonstrated that patients identified at high risk of malnutrition by the Nutritional Risk Screening (NRS) risk system benefit from an integrated intervention [9].

Line 338-9: ... and potentially integrated into routine care practices and lead to reduced hospital length of stay, and better patient outcomes.

Suggestions from reviewer were taken. Sentence was rephrased accordantly.

Line 343 omit the ie from lack of a ....

Suggestions from reviewer were taken. Sentence was rephrased accordantly.

Lines 345-6 Also used twice in 1 sentence  

Suggestions from reviewer were taken. Sentence was rephrased with:

Educational gaps of healthcare professionals in the field of nutrition, as well as the importance given to nutrition care might also be considered crucial to fight DRM.

Line 351: ASPERN requires further efforts

Suggestions from reviewer were taken. Sentence was rephrased accordantly.

Line 358: what does proper mean here? Suggest replacing with a more appropriate word or expand to clarify.

Section 4.2 was removed from manuscript for not being sufficiently supported by our data.

Line 361: Do you mean nutrients? Perhaps foods?

Section 4.2 was removed from manuscript for not being sufficiently supported by our data.

Page 12, final box in LHC Missing a partnership for ... 

Suggestions from reviewer were taken. Sentence was rephrased accordantly.

RHC, box 2: ....homes and also target... Final sentence: ....education platform ...

Suggestions from reviewer were taken. Sentence was rephrased accordantly. 

RHC, final box: Support for use of? Supported use of local food in hospital kitchen

Page 13, line 376: In summary the analysis... and provides discussion for potential

Suggestions from reviewer were taken. Sentence was rephrased accordantly.

Line 379: ...DRM to all...

Suggestions from reviewer were taken. Sentence was rephrased accordantly.

Line 383: DRM is unlikely to disappear from hospital related disease acting as a strong driver of malnutrition.

Suggestions from reviewer were taken. Sentence was rephrased accordantly.

Line 384: Appropriate nutrition care should be ...

Suggestions from reviewer were taken. Sentence was rephrased accordantly.

Reviewer 2 Report

The authors should be commended for attempting such a large effort to bring heightened awareness to nutrition issues. The concept is interesting but I have several methodological questions: 

Do I understand correctly that all of the data was collected on one day, thereby providing a "snapshot" of a patients' nutritional status on that day? It is problematic to tie 30-day outcomes to the quality of nutrition on one day, not taking into account the patient may have eaten a full diet the day before or the day after nutritionDay. Or conversely, may have had a full diet on nutritionDay but spent the next several days not eating. (Figure 3). However, the data regarding weight change and meals eaten the week prior provide a more longitudinal view of the patients' status and are a more compelling argument to be tied to the outcome of death.  How do you account for disease severity or  comorbid conditions? How can you be certain that a patient not eating on nutritionDay is associated with death when there is no measure of severity of comorbid conditions such as severe cardiopulmonary disease, renal disease, cancer, etc. Do you have data on APACHE scores, a comorbidity index (Charlson or similar) ? Table 2 is difficult to read. Recommend revision and highlighting notable findings or notable differences between groups. Figure 2 is also difficult to read. What is the take home message?  Table 3. The data presented reveals the discharge disposition of medical patients. But as stated before, the one day snapshot of nutrition cannot be correlated with discharge disposition. In light of that limitation, what is the message to the reader with Table 3?  The data you present regarding mortality and nutrition risk factors is the most compelling and affirms other studies associating nutritional risk with poor outcomes. I would leverage this data more than the data about the nutritional intake on one random hospital day.  Figure 3 contains a large amount of data and is difficult to read. Again, fluid status and nutrition intake on one day cannot be correlated to death. The data is also confounded by the heterogeneity of the groups. Patients who are not mobile on nutrition day may have severe frailty, or they may be limited by surgical drains. Those are two very different risk factors for death and are unaccounted for in your statistical analysis.  How do you define fluid status of normal, overload and dry? Please provide definitions.  Section 4.2 is interesting but is not supported by the data you present.  Similarly, in your conclusions, you discuss reimbursement and responsibilities regarding nutrition delivery, but this is not supported by the data you presented.  Your conclusion that a systematic approach to DRM and continuity of care may reduce hospital length of stay is not substantiated by your data, nor do you present cost data. It is an overstatement to present this conclusion. 

The concept of auditing nutrition history, orders, and monitoring is a noble one and the increased awareness that can be generated from a study of this size is indeed impressive. However, I would caution the use of the audit (especially fluid status and diet orders) data to comment on outcomes such as mortality. Consider refocusing the study on the the gap in recognizing malnutrition, recording nutrition data, etc. Figure 1 captures this data well and is easier to read.

Author Response

Reviewer 2:  

The authors should be commended for attempting such a large effort to bring heightened awareness to nutrition issues. The concept is interesting but I have several methodological questions: 

Do I understand correctly that all of the data was collected on one day, thereby providing a "snapshot" of a patients' nutritional status on that day? It is problematic to tie 30-days outcomes to the quality of nutrition on one day, not taking into account the patient may have eaten a full diet the day before or the day after nutritionDay. Or conversely, may have had a full diet on nutritionDay but spent the next several days not eating.

Response: Yes, nutritionDay represents patients' nutritional status on that day enriched with information based on questions addressing the patient’s nutritional history such as “Did your weight change?” or “How much did you eat last week?”.

We agree with the reviewer that patients may eat the full meal the day before or the day after nutritionDay and that the one day snap-shot may not represent food intake for the full hospital stay. Nevertheless, if we consider “food intake on a given day (nutritionDay)” as a clinical“symptom”among many others, we can attempt to analyze the “symptom” for association with outcome. If such a symptom can be shown to be associated with a “hard” outcome such as death in hospital in univariate analysis (after adjustment for multiple risk factors) we consider that the symptom is worth clinical attention. In fact, low food intake on nutritionDay was associated with death in the year 2006, in theoverall cohort with HR up to 2.71 [1.88-3.91] for eating a quarter or nothing. (Clinical Nutrition 2009; 28: 484-491). In an another analysis that led to the development of  the PANDORA score for prediction of death in hospital within 30 days after nutritionDay, again decreased food intake was identified among the seven most important risk factors as it contritbuted 3-12 points out of a maximum of 75 points to the final score . Only 7 variables out of 52 with 123 item classes were finally included in the score (PLoS ONE 2016; 10(5): e0127316).   

 (Figure 1). Thus we think that eating in hospital on a given day is an important risk indicator as they are “eating in the week before admission and “eating loss in the previous three months”.

We have added in the manuscript text the following:

In the nutritionDay analysis, several nutrition related risk factors such as low BMI, recent weight loss and reduced food intake (in the week before nutritionDay or on nutritionDay itself) were found to be independently associated with death within 30 days in hospital. In an another analysis that led to the development of the PANDORA score for prediction of death in hospital within 30 days after nutritionDay, decreased food intake was identified among the seven most important risk factors as it contritbuted 3-12 points out of a maximum of 75 points to the final score.   ….

However, the data regarding weight change and meals eaten the week prior provide a more longitudinal view of the patients' status and are a more compelling argument to be tied to the outcome of death. 

Response: We agree and in attempt to identify the proper weight of variables in the multivariate analysis for 4 patient subgroups (medical, surgical, longterm care and other specialties)..

How do you account for disease severity or  comorbid conditions?

How can you be certain that a patient not eating on nutritionDay is associated with death when there is no measure of severity of comorbid conditions such as severe cardiopulmonary disease, renal disease, cancer, etc. Do you have data on APACHE scores, a comorbidity index (Charlson or similar)?

Response: We have included in the multivariate analysis in addition all diagnostic categories referring to the affected organs based on ICD10 categories and all six comorbidities that are recorded in the nutritionDay database to account for disease specific factors and organ related severity of illness. In this regard, figure 4 was updated and figure 5 was added to highlight the above described adjustment.

We added this sentence as well into the experimental section:

In addition we included in the multivariate analysis the diagnostic categories derived from the 17 ICD 10 top categories andsix comorbidities (diabetes, stroke, COPD, myocardial infarction, cardiac failure and others).  

Table 2 is difficult to read. Recommend revision and highlighting notable findings or notable differences between groups.

Response: We agree with the reviewer and we reduced its content by omitting some redundant categories not to generate confusion and to get the reader focused only on  the most important points.

We provided further explanations at the bottom of the table:

Multiple entries for one patient possible. Patients who ate nothing on nutritionDay were divided in those who did not eat even if they were allowed to (nothing-eating allowed) and those who did not eat because they were told not to eat by the doctor (nothing-eating not allowed) as for instance before a planned surgery or diagnostic tests. Combined enteral and parenteral nutrition was used in less than 2.5% of patients in any patient category and was omitted.

Figure 2 is also difficult to read. What is the take home message?

Response: We agree with the reviewer and we have made major changes to Figure 2. We first focused the attention on describing the use of nutritional care process, such as screening, in the participating units (cohort 2016-18). We then tried to describe the proportion of identified malnourished/at risk patients with the screening used.

We added the following text to the manuscript:

A total of 1415 units reported in 2016-2018 about the screening tool utilized (Figure 2). The majority used a formal tool such as NRS-2002, MUST, MST or a local tool. Only 10% did not have a routine screening nor fixed screening criteria. Patients identified as malnourished were overall 3340/28100 (11.9%), with the  highest proportion identified with an informal tool 722/4793 (15.1%) or NRS-2002 1169/8453 (13.8%) followed by MUST 134/1210 (11.1%), MST 269/2269 (11.9%) and visual experience 197/1666 (11.8%) whereas the proportion of identified malnourished patients was lower with no implemented routine 204/2154 (9.5%). The proportion of patients identified at nutritional risk was overall 4944/28100 (17.6%). Of those, the highest proportion, 22.4% was identified with an unspecified tool while only 14-16% of patients were identified at risk when no routine, only visual appearance or MST was used. Only 28.9% of malnourished patients had a BMI below 18.5, 49.8% a normal BMI, 21.4% were overweight or obese. Being identified as malnourished was associated with unintentional weight loss within the last three months in 85% of patients, with reduced nutrient intake before admission in 57% of patients and with not eating a full lunch in hospital in 65% of patients. Not being identified as malnourished or at risk of malnutrition was associated with unintentional weight loss within the last three months in 47% of patients, with reduced nutrient intake before admission in 28% of patients and with not eating a full lunch in hospital in 48% of patients. The proportion of patients with two or more nutrition risk factors was 50% in the cohort 2016-2018 comparing normal as reference and all patients with low BMI, weight loss and low intake as at risk.

Secondly, we added a new figure 3, which continues to describe the use of nutritional care processes within the units such as “nutrition intake monitoring” and “nutrition expert consultation”. Those two are indicators of an attentive and continuous nutritional care in the unit as they represent following steps post screening. We added/modified text in the manuscript as following: Figure 3: Nutrition care process indicators versus three nutrition associated risk factors. Daily nutrition intake monitoring (left) and nutrition expert consulted (right). Bars indicate percentage answering “yes”. Missing values were < 7.5% in all subcategories. Color coding similar to figure 1 and given on x-axis (empty bar is reference).

Nutrition intake monitoring was more frequent in patients with unintentional weight loss than in patients with stable weight (52% versus 41%) (Figure 2 ). Still, nearly 50% of patients with a history of weight loss did not have their nutrition intake monitored while in hospital. Similarly, history of poor nutrition intake before admission triggered more frequent monitoring compared with history of normal eating but intake was never monitored in more than 50% of patients with a history of poor nutritional intake. Actual poor food intake did not have any effect on monitoring of food intake.

A nutrition expert was consulted more frequently when an unintentional weight loss was reported by patients or when food intake was reported as reduced before hospital admission (Figure 2). A nutrition expert was never consulted in more than 46% of patients even when risk factors were reported. Malnutrition reporting followed a similar pattern as consulting an expert and never was above 41%.

Low BMI was associated with a more frequent monitoring of intake (58% versus 47%) and malnutrition reporting in patient record (52% versus 36%) compared with normal BMI.  In obese patients (BMI 30-35) we observed the lowest intake monitoring (40%) and malnutrition reporting (27%).

Table 3. The data presented reveals the discharge disposition of medical patients. But as stated before, the one day snapshot of nutrition cannot be correlated with discharge disposition. In light of that limitation, what is the message to the reader with Table 3?

Response:  table 3 has been reduced in complexity by omission of the sensitivity analysis present in the last two columns. Reporting outcome of the interviewed patients is a central part of the nutritionDay study design and it is an important descriptor of the cohort studied.

The data you present regarding mortality and nutrition risk factors is the most compelling and affirms other studies associating nutritional risk with poor outcomes. I would leverage this data more than the data about the nutritional intake on one random hospital day.

Response:  thank you for your comment, please see the above answer on risk factors and outcome (death).

Figure 3 contains a large amount of data and is difficult to read. Again, fluid status and nutrition intake on one day cannot be correlated to death. The data is also confounded by the heterogeneity of the groups. Patients who are not mobile on nutritionday may have severe frailty, or they may be limited by surgical drains. Those are two very different risk factors for death and are unaccounted for in your statistical analysis.

Response: We have redone the multivariate analysis to address the heterogeneity of the groups and includes now diagnostic categories and comorbidities. Those two aspects are now shown in figure 4 and figure 5 respectively. The pattern of association between risk factors and death in hospital 30 days after nutritionDay has been confirmed by the new multivariate analysis. The mobility differences between surgical and medical patients is adequately addressed by the analysis and shows that reduced mobility has a larger effect in medical patient but reduced mobility remains a risk factor in both groups (fig 4).  

How do you define fluid status of normal, overload and dry? Please provide definitions. 

Response: We added in the experimental section a definition: fluid status (as intravascular or tissue fluid overload or depletion as observed by a clinician)

Section 4.2 is interesting but is not supported by the data you present.  Similarly, in your conclusions, you discuss reimbursement and responsibilities regarding nutrition delivery, but this is not supported by the data you presented. 

Response: We agree with the reviewer that section 4.2 about food provision as well as reimbursement schemes in nutrition delivery in the conclusions are not fully supported by the data we presented and we have deleted them from the manuscript.

Your conclusion that a systematic approach to DRM and continuity of care may reduce hospital length of stay is not substantiated by your data, nor do you present cost data. It is an overstatement to present this conclusion.

Response: We agree with the reviewer that length of stay and costs are not part of this analysis and we have changed the sentence to:

A systematic approach to DRM in hospitals and an adequate continuity of care  may lead to better outcomes. nutritionDay in this regard has served and it will keep serving as a tool to monitor changes in clinical practice and associated outcome.

The concept of auditing nutrition history, orders, and monitoring is a noble one and the increased awareness that can be generated from a study of this size is indeed impressive. However, I would caution the use of the audit (especially fluid status and diet orders) data to comment on outcomes such as mortality. Consider refocusing the study on the the gap in recognizing malnutrition, recording nutrition data, etc. Figure 1 captures this data well and is easier to read.

We would like to thank reviewer 2 for his valuable scientific comments that allowed us to improve readability and clarity of the manuscript.

Round 2

Reviewer 2 Report

The authors should be commended for their tremendous work and their detailed responses to my initial questions. I appreciate their thoughtful replies and the additions, deletions, or corrections they provided. 

I do have several additional questions/concerns: 

Introduction, p.2. line 69-70: please clarify the reference used to support the statement "Patients with malnutritionusually stay longer in hospitals..." Does reference 9, 10, 11 support this, which is noted several sentences later.  Methods: p. 4 Why did you not exlude patients with missing data? Your sample size of 59,046 would leave you with an impressive number of patients in your analysis without having the extra columns, boxes, lines and rows in many of your tables. The patients with missing data introduces confusion into the analysis.  p. 4 lines 172-177, you describe the four groups. Why were surgical specialty patients not included in the surgery group (trauma, gynecology, orthopedics, etc)? Why do you describe these groups but provide analysis for medical and surgical patients in the tables? It appears then, that, surgical speciality patients (trauma, gynecology, etc) are not included in the multivariate analysis (Fig 4 and 5). Can you comment on these other groups of patients? The non-medical, non-surgical patients appear to account for 20% of the total. Do you have data to include them in the final analysis?  p.4 line 178-179: Do you have more objective data regarding fluid status besides "as observed by a clinician?, which is a highly subjective measure.  p. 4 line 182-183, please further clarify the comorbid conditions and organs affected (as noted in Figure 5). Why was cancer not included in the co-morbid conditions? Cancer would be a strong predictor of malnutrition related morbidity and mortality.  p. 4, comorbid conditions: how do you account for patients who have multiple comorbid conditions? Was this accounted for your multivariate analysis? If so, please clarify.  Table 2 remains hard to interpret. Please either reformat or provide additional narrative. It's unclear what the percentages mean when reading across the table. If this table represents only medical patients, do you have similar analysis of surgical and other groups?  Figure 3. Do you have statistical analysis to determine significant differences between the groups? From the figure, the bars appear very similar in height, what is statistically different and what is not?  p.9 line 296-299 Do you have statistical analysis to express signifcant differences in addition to the percentages?  If patients with missing data are removed, Table 3 would become easier to read.  Figures 4 and 5: remain busy and difficult to read. Consider highlighting or color coding the significant findings.  Figure 5: please further define what is meant by "affected organs" as I didn't see this clearly defined in the background or methods; what if a patient has multiple organs affected? How is this accounted for?  Take caution not to overstate your conclusions: p. 12 lines 364-366 "Risk factors such as low BMI...associated with poor hospital outcome within 30 days of nutritionDay." Be careful not to insinuate that it's the malnutrition that caused the morbidity and mortality, as you stated earlier in the paper that malnutrition awareness if often thwarted by the fact that short term (30 day) outcomes are frequently not directly impacted by bouts of suboptimal nutrition. It may be prudent to specifically states that malnutrition and risk factors are prevelant, and that nutrition care processes are not concordant with the degree of malnutrition noted at admission, but that morbidity and mortality are heavily driven by disease processes (cancer, trauma, respiratory failure, etc) and that the observation in the study is simply that, observation, correlation, not causation. You don't assert causation, but it would be wise to explictity state only correlation is noted.  Please add a limitations section. There are several limitations in the use of large databanks and these should be mentioned. There is missing data, loss of granularity, heterogenity. Taking mass quanitities of data and analyzing in large only moderately similar groups of patients carries the risk of overstating claims. This should be mentioned.  Please revise your conclusion statement to summarize the findings as presented in your results section. The conclusions discuss national policies, cost data, quality of life, patient autonomy and several other topics not addressed by your data. 

Author Response

Response to review:

Thanks again for all careful reading and suggestions fro improvements.

The authors should be commended for their tremendous work and their detailed responses to my initial questions. I appreciate their thoughtful replies and the additions, deletions, or corrections they provided. 

I do have several additional questions/concerns: 

Introduction, p.2. line 69-70: please clarify the reference used to support the statement "Patients with malnutrition usually stay longer in hospitals..." Does reference 9, 10, 11 support this, which is noted several sentences later. 

We have noticed the formatting mistake. References 9 and 10 have been moved to line 69-7. We have added two more references on the topic.

Methods: p. 4 Why did you not exclude patients with missing data? Your sample size of 59,046 would leave you with an impressive number of patients in your analysis without having the extra columns, boxes, lines and rows in many of your tables. The patients with missing data introduces confusion into the analysis. 

The nDay study has a very complex structure of missing values. Unit participation as well as patients’ participation to the study is voluntary and it also requires a direct feedback from the patients themselves. Therefore a certain amount of missing data can be expected especially in such a big international study.

Sometimes one question can be overlooked, unknown, too sensitive, or unable to be answered. Excluding a missing value in a given variable would mean to exclude all given/valid answers of that patient from analysis.

For sensitivity we have run the complete cases analysis as well and found similar results. No significant risk indicator did change direction and a few indicators even improved in strength of association. One example would be the association with better outcome in surgical patients if overweight or obese. In the complete case analysis there 82993 cases compared with the actual analysis based on a total of 153470 cases. With losing about 46% of data for analysis, potentially creating bias and in accordance to the STROBE guidelines we decided to show the full analysis with the missing data as main result. We have therefore shown the % of missing within that variable not to lose precious information related to other variables.

We have addressed this sensitivity analysis in the methods.

4 lines 172-177, you describe the four groups. Why were surgical specialty patients not included in the surgery group (trauma, gynecology, orthopedics, etc)?

In the group „surgical“ were included patients admitted to a surgical unit or waiting for surgery or after surgery on any medical/other ward.

This is clarified by showing in Table 2 all 4 groups. Thus all patients with a planned or performed surgical intervention from trauma, gynecology, ect. are included in the surgical group.

Why do you describe these groups but provide analysis for medical and surgical patients in the tables? It appears then, that, surgical specialty patients (trauma, gynecology, etc) are not included in the multivariate analysis (Fig 4 and 5). Can you comment on these other groups of patients? The non-medical, non-surgical patients appear to account for 20% of the total. Do you have data to include them in the final analysis? 

The multivariate analysis was run in the four groups (FIG 4) and we have highlighted significant differences in color and linetype coded form to improve readability. The non-medical, non-surgical patients groups contain enough data to compare the multivariate analysis.

p.4 line 178-179: Do you have more objective data regarding fluid status besides "as observed by a clinician?, which is a highly subjective measure. 

In both dataset analyzed, fluid status was addressed with only one question directed to  the caregivers of the participating unit. In the nutritionDay study, questions are designed to be easy and fast to reply, and the experience of the unit staff was accepted to distinguish between a normal overloaded or dehydrated status.

We agree with you that fluid status is a subjective classification where individual experience and the interpretation of the patient’s actual situation may enter. Nevertheless we found in all multivariate analysis that the association of this “symptom” with death after 30 days in hospital was very consistent and robust. In the actual analysis this is shown in all 4 groups. In normal wards it is common practice to examine a patient and take the history and give all these elements a weight in the general patient evaluation.

4 line 182-183, please further clarify the comorbid conditions and organs affected (as noted in Figure 5). Why was cancer not included in the co-morbid conditions? Cancer would be a strong predictor of malnutrition related morbidity and mortality. 

In the questionnaires, cancer appear among the options of affected organs (in the 2006-15 dataset). Cancer was included in the model but it was just not labeled as comorbidity but as an affected organ. As shown in figure 5, cancer is indeed a strong predictor of malnutrition.

4, comorbid conditions: how do you account for patients who have multiple comorbid conditions? Was this accounted for your multivariate analysis? If so, please clarify. 

All diseased organs as well as the comorbidities are included in the model as unique variables, this means „YES - having that diseased organ/comorbidities” or „NO, not having that diseased organ/comorbidity“. This means that each patient has his own profile of diseased organs and comorbidities included in the model. In this way, we can account for the presence of multiple conditions at the same time. Moreover, we have included the following sentence in the experimental section to further clarify (line 182:186):

In addition we included in the multivariate analysis the diagnostic categories derived from the 17 ICD 10 top categories (brain and nerves, eye and ear, nose and throat, heart and circulation, lung, liver, gastrointestinal tract, kidney/urinary tract/female genital tract, endocrine system, skeleton/bone/muscle, blood/bone marrow, skin, ischemia, cancer, infection, pregnancy, others) and six comorbidities (diabetes, stroke, COPD, myocardial infarction, cardiac failure and others), each one used as a unique variable as the condition being present vs not present

Table 2 remains hard to interpret. Please either reformat or provide additional narrative. It's unclear what the percentages mean when reading across the table. If this table represents only medical patients, do you have similar analysis of surgical and other groups? 

We have rearranged the layout of the table to make comparison between categories easier and we have also merged N and percentage into the same line. COMMENT: Table 2 is now labeled as Table 3.

Figure 3. Do you have statistical analysis to determine significant differences between the groups? From the figure, the bars appear very similar in height, what is statistically different and what is not? 

We added the statistical comparison to each reference group to the plot. Also added on sentence within the experimental section (line 201:204). You precisely noted that the differences between bars appear small but because the number of observations is so large many categories are significantly different from the reference group. We have corrected for multiple testing and do only mark highly significant (p<0.005 or higher) differences. Moreover we are always very conservative in the interpretation of small differences even if significant.

p.9 line 296-299 Do you have statistical analysis to express significant differences in addition to the percentages? 

If patients with missing data are removed, Table 3 would become easier to read. 

See answer about missing values.

Figures 4 and 5: remain busy and difficult to read. Consider highlighting or color coding the significant findings. 

We have highlighted significant differences in color and line-type coded form to improve readability.

Figure 5: please further define what is meant by "affected organs" as I didn't see this clearly defined in the background or methods; what if a patient has multiple organs affected? How is this accounted for? 

See answer about multiple comorbidities. We also renamed the affected organs to diseased organs to add clarity

Take caution not to overstate your conclusions: p. 12 lines 364-366 "Risk factors such as low BMI...associated with poor hospital outcome within 30 days of nutritionDay." Be careful not to insinuate that it's the malnutrition that caused the morbidity and mortality, as you stated earlier in the paper that malnutrition awareness if often thwarted by the fact that short term (30 day) outcomes are frequently not directly impacted by bouts of suboptimal nutrition.

We have renamed “risk factors” to “risk indicators” and have replaced “increased” by “higher” in order to avoid suggesting a causal association that cannot be deduced from cross-sectional observational data.

It may be prudent to specifically states that malnutrition and risk factors are prevelant, and that nutrition care processes are not concordant with the degree of malnutrition noted at admission, but that morbidity and mortality are heavily driven by disease processes (cancer, trauma, respiratory failure, etc) and that the observation in the study is simply that, observation, correlation, not causation. You don't assert causation, but it would be wise to explictity state only correlation is noted. 

We have added now for all groups the results and include the effect of organs into the models. Of course there are certainly different categories within an ICD organ group that are very different from each other such as myocardial infarction or heart failure. We aim only at identifying the most robust risk indicators that may serve as trigger for improved care.

Please add a limitations section. There are several limitations in the use of large databanks and these should be mentioned. There is missing data, loss of granularity, heterogenity.

A limitation section that is clearly identifiable has been included and addresses issues such as missingness and unprecise clinical observations.

Taking mass quanitities of data and analyzing in large only moderately similar groups of patients carries the risk of overstating claims. This should be mentioned. 

We have taken great against overstating associations and hope that the diversity of groups partially addressed by the multivariate analysis.

 Please revise your conclusion statement to summarize the findings as presented in your results section. The conclusions discuss national policies, cost data, quality of life, patient autonomy and several other topics not addressed by your data.

We feel that such a large data collection and the critical analysis that we have done with your suggestions from the reviewing process should allow to give some thoughts on further directions that may be taken to improve or study DRM.
